# Assessment of School Food Policy Influencing Nutritional Behaviour of Adolescents from the Perspective of School Stakeholders in Ibadan, Oyo State

**DOI:** 10.3390/ijerph22060866

**Published:** 2025-05-31

**Authors:** Mary Ibukunoluwa Tubi, Oyediran Emmanuel Oyewole

**Affiliations:** 1Department of Nutrition and Dietetics, Bowen University, Iwo 232102, Nigeria; 2Department of Health Promotion and Education, University of Ibadan, Ibadan 200005, Nigeria; o.e.oyewole@com.ui.edu.ng

**Keywords:** school food policy, nutritional behaviour, key informant, principal, teacher, food service staff

## Abstract

School food policy (SFP) that promotes nutritional behaviour is a national priority. Despite the role of school principals, teachers, and food service staff (FSS) in implementing SFP, their contribution to the current state of school food policy remains unexplored. The study aims to explore the perspective of these stakeholders on the effectiveness of SFP, barriers, and future recommendations for effective implementation. A qualitative study using 24 key informants interviews including principals (*n* = 6), teachers (*n* = 12), and FSS (*n* = 6) was conducted. From the list of private secondary schools, six schools were randomly selected. Principals, teachers, and FSS were selected through purposive sampling. A thematic approach was adopted for the data analysis. Interview data were categorised into three broad themes: (1) availability of school food policy, (2) barriers to enforcement, and (3) recommendations. Most participants stated that policy implementation is not enforced. The barriers are non-prioritised funding, resistance to change, and time constraints. The recommendations include regular training, activity manual development, seasonal food promotion, nutrition education, and leadership support. The study provides a deeper understanding of the role of key stakeholders in the current state of the effectiveness of SFP implementation. The inclusion of stakeholders is essential for a successful school food policy.

## 1. Introduction

School food policy that promotes nutritional behaviour among adolescents is a priority [1,2]. However, successful implementation remains a challenge. According to studies, energy-dense, nutrient-deficient foods are becoming more readily available and accessible in school settings globally [3,4] as well as in Nigeria [5,6] in addition to the poor availability of nutrient-dense foods [7]. According to studies, the majority of the food available at school tuck shops is energy-dense, nutrient-poor, and sugar-sweetened, further encouraging young individuals to consume these unhealthy food items [8,9]. This condition indicates poor access to nutrient-dense meals in many school canteens and tuck shops, which discourages adolescents from making wise food decisions [10].

Additionally, adolescents from the higher and middle socioeconomic tiers and attending private school are a distinct population often with different resources compared to public schools, which can potentially impact students’ nutritional behaviour. [11], studied the weight status and eating habits of 2097 urban teenage school girls from middle-income families and found that 1009 (48.1%) acknowledged skipping at least one meal every two weeks. More than half of the participants (60.2%) ate fast food at least once a week with more than three-quarters (76.4%) of them ordering fast food and soft beverages. According to another survey, the majority of obese boys were from high-income families. When it comes to girls, the majority of obese girls came from intermediate-income homes, while the overweight girls came from high-income families. This revealed that high and intermediate socioeconomic characteristics are a major risk factor for adolescent overweight and obesity [12,13,14]. The majority of these adolescents come from families with average to high incomes that can afford a variety of meals and snacks and are at high risk of making poor dietary choices. However, it is challenging to encourage students to adopt positive nutritional behaviour given the accessibility of foods that are high in calories but low in nutrients, the scarcity of nutrient-dense foods, the ineffective implementation of written school food policies, and the high cost of nutrient-dense foods in the educational setting [15]. These have devastating effects, especially because they are at a stage of developing autonomy in decision making such as in the choice of food and related lifestyle [16].

Nutritional behaviour is therefore of high significance at this stage, because adequate nutrition is vital to meet the requirements of rapid growth and development, increased physiological activities, cognitive growth and development, and the prevention of illnesses and nutrition-related diseases [17]. When nutrition demands are not met, adolescents become malnourished, which consequently has devastating effects on their development into productive adults. Several studies have reported the burden of malnutrition among adolescents. In a cross-sectional survey of 172 pupils in private schools in Port Harcourt, aged 10 to 16, 47% were classified as normal weight, 46.2% as underweight, and 6.6% as overweight [18]. Another study among adolescents in southwest Nigeria showed malnutrition among in-school adolescents, where nearly one third were underweight and 7.6% were overweight [19]. Similar prevalence shows underweight, overweight, and stunting percentages of 39.3%, 8.0%, and 2.5%, respectively, in a private secondary school in Oyo State, Nigeria [20].

School nutrition policies ought to limit the availability of less healthy meals and beverages, according to the World Health Organisation; the school implements the food policy, which is regulated by the government [21]. It is also important to note that the availability, affordability, and accessibility of unhealthy meals in school environments contribute to adversely influencing nutritional behaviour [22]. Also, school food policies have been successfully implemented in developed countries [23,24] as an intervention to reduce malnutrition. In addition, Nigeria introduced a national school feeding programme in 2016 to provide adequate and nutritious meals to all primary school pupils [1,25]. The programme has been reported to take place in 26 states around the country, involving 80,000 farmers and over 102,097 cooks with over nine million children benefiting from 54,619 schools [26]. Studies have shown that participants are aware of the existence of the programme but have reported poor implementation [1,27]. These studies have shown a poor quality and quantity of meals provided for the pupils. Also, the school food policy emphasises incorporating nutrition education and training into curricula and encouraging healthy eating practices. These have also faced challenges in effective implementation, thereby contributing to the increase in poor nutritional behaviour among adolescents. Some of the challenges resulting in poor implementation as reported include inadequate resources, poor monitoring and poor quality of school meals, and limited training opportunities [1,27,28,29]. 

School food policy implementation is enabled by the key stakeholders in the school, including the school principals, teachers, and food service staff [27,30]. These stakeholders play an important role in developing and enforcing school food policy. Their opinions and experiences provide an understanding of critical insights into the practical aspects of policy implementation and monitoring as it influences adolescent nutritional behaviour [30]. Despite the implementation in primary schools in Nigeria, there is no known study on the role of the principal, teachers, and food service staff also known as the staff at the food tuck shop, who are the key stakeholders in the implementation and enforcement of policy implementation among adolescents, especially those in private secondary schools in Nigeria. 

Therefore, to promote the positive adoption of the school food policy by adolescents, an examination of the role of these key stakeholders in promoting and adhering to the policies must be considered. The understanding will provide valuable information about the strengths and weaknesses of present policies implementation in secondary school as well as the contextual elements that determine their efficacy. It will also provide valuable insight into ensuring that clear and strict school food policies that ensure a better availability of healthier food options and reduced accessibility to unhealthy foods are implemented. It is likewise important to explore how these policies are communicated, monitored, and enforced by these key stakeholders within the school setting. 

The study’s major findings will present an overview of the current situation of school food policy implementation, exploring the availability of healthy eating policies. It will also highlight how these policies are communicated, monitored, and enforced within the school setting by examining the role of the school administrators, teachers, and food service staff in promoting and adhering to the policies. The opinions of major stakeholders can shed insight into the challenges of implementing food policies and aid in the identification of potential solutions. These findings can be used to guide future interventions and policy changes to promote the proper implementation of school food policies, thereby promoting healthier nutritional behaviour among secondary school students. Therefore, the purpose of this study is to understand the perspectives of school principals, teachers, and food service staff as related to the school food policies’ implementation and enforcement in school. 

## 2. Materials and Methods

This qualitative study utilised a thematic analysis. From June to August 2022, 24 in-depth interviews were conducted with key informants among three subgroups, as shown in Table 1. Stratified purposive sampling was used to select the study areas. Five research assistants who were graduates with qualitative research experience, two from the field of public health and three from the Department of Human Nutrition and Dietetics, were recruited for the study. The Research Assistants further received a two-day training. The research team was instructed to use a training manual that was designed specifically for the training activity. The training included the basics of the research principles, the importance of objectivity and unbias, the research objectives, and lastly, how to implement KII and FGD sessions. Purposive sampling was used in recruiting key informants from six private English-speaking secondary schools in Ibadan, Oyo State. Private school students are mostly from the high or middle-income class with reported cases of non-communicable diseases [31]. According to Ahmad, A et colab. (2018) [12], adolescents in this socioeconomic tier had a greater prevalence of obesity and overweight. Therefore, targeted intervention for this group of young people will aim at limiting the grievous consequences of non-communicable diseases. From the list of private secondary schools, six schools were randomly selected. Location includes Sango, Oluyole, Bodija, Mokola, Agbowo, and Total Garden. All are within the Ibadan metropolis, Oyo State.

Participants were selected because they are the primary stakeholders involved in decision making related to school food policy implementation. They are also expected to be well informed about the school food environment and school food policies. Interviews included six with principals, one each from the six (*n* = 6) identified schools, twelve interviews (*n* = 12) with school teachers (one home economics teacher and one food and nutrition teacher from each school) and six interviews with food service staff, one each from the six (*n* = 6) identified schools. The key informants were actively involved in affairs relating to food within the school. 

In order to explore the opinion of these key stakeholders in school food policy implementation, a key informant guide was developed by the research team. It sought to understand the effectiveness of existing food policies, identify barriers and challenges to implementation, and provide recommendations for future interventions. Their opinions were recorded and analysed. Conducting the interviews, analysing the data, and reporting were in accordance with the Consolidated Criteria for Reporting Qualitative Research (COREQ) requirements [32].

The modified interview guide used was created by the United States Agency for International Development (USAID) Policy Implementation Assessment Tool for programme implementers and other stakeholders. The interview guide comprised open-ended questions and was applied to the analysis of health policy and programmes in numerous low- and middle-income nations. The policy implementation framework by the USAID was used to demonstrate the connections between health-related programme execution, policy making, and health outcomes [33]. The framework was used to identify themes and also data patterns. The study team tested and refined the question guide once it had been developed, and it was also pilot-tested with two principals, two food subject teachers, and two school food service staff. The question guide included an introduction and opening questions that enabled the informant to establish a sense of familiarity before diving into the topic at hand. To focus the attention on the goal of this study in understanding the factors that influence successful school food policy implementation, transition and important questions were employed, respectively. Three researchers trained in qualitative interviewing conducted the interviews. Two of the researchers held masters in human nutrition degrees, and the other researcher held a master’s degree in public health.

The study complied with the ethical standards that guide research with human subjects. The Oyo State Research Ethical Review Committee, Ministry of Health, Secretariat, Ibadan granted the ethical approval with reference number AD 13/479:1647^A^. The study’s objectives were extensively explained in English, and participants were made aware that taking part in the study was completely optional. All ethical concerns were carefully handled. Consent was also obtained from the adolescents, and consent from their parents was granted before their enrolment.

The sessions were held in suitable areas within the school to avoid distractions or noise and also at a convenient time for the participants. Each session lasted 45–60 min on average. The discussion was led by a moderator, while someone took notes and documented the main points of the discussion. The purpose and objectives of the study were explained to all the participants. The confidentiality of the information shared was assured, and their willingness to participate and informed consent were subsequently confirmed. The researcher ensured saturation was achieved across the interviewees by checking for no new data and redundant data for each of the specific questions asked. After approximately 24 interviews, the researcher determined that saturation occurred, and thus no additional interviews were needed.

The discussants requested approval to utilise a digitised recording device before the KII sessions started. All identifiers were erased, including participant names. The numbers assigned to each person were not connected to any identifiers and were just used to facilitate discussion. Both verbal and nonverbal communications were also recorded by an observer. Yin’s five-phase analysis method, which entails compiling, disassembling, reassembling, interpreting, and concluding, was employed in analysing the data [34].

In the compiling phase, interviews were transcribed. All of the transcripts from the 24 key informants were then given individual code numbers for principals (P1 to P6), teachers (T1 to T12), and food service staff (F1 to F6). 

The disassembly phase comprised reading the transcript, repeatedly listening to interviews, and observing trends in data. Following this, precise data segments that matched the study’s objectives were identified and assigned labels to the data segments to generate preliminary themes, codes, and subcodes based on the conceptual framework provided by the USAID. The set of codes and subcodes that recurred predictably were categorised into a theme. All quotes were encoded using the qualitative software program ATLAS Ti version 12. Using a thematic approach, data (quotes) were examined for themes. 

In the reassembling stage, similar quotes were grouped into more general concepts (subthemes) and further categorised into main themes. The outputs for each instrument were generated and merged after all of the interviews were coded. Based on the themes from the purpose of the study, thematic analyses of the themes that had been identified were carried out. To ensure the reliability of data interpretations, analyses were carried out independently by two researchers. 

In the interpretation phase, summarises were prepared to interpret the data and also discuss significant quotations. Finally, conclusions were drawn from the data.

## 3. Result

Three themes emerged from the interviews: (1) availability of school food policy, (2) barriers to policy enforcement, and (3) recommendations for successful future implementation of school food policy. Result is presented as shown in Table 2. 

### 3.1. Theme 1: Availability of School Food Policy

#### 3.1.1. A Policy That Restricts the Availability of Unhealthy Foods

The majority of stakeholders noted that the school does not have a written school food policy. The few schools with school food policies do not have them enforced. They also noted that the school management does not regulate food sold and brought into the schools. Also, all sorts of snacks were sold in the school store. 


*“We do not have a written food policy in the school. Also, cooked foods are not often sold in school except occasionally, when some vendors bringing in seasonal vegetables such as corn and mangoes”*
(F3)


*“We have a written policy on the availability of healthy foods in school, but it is not enforced. Also, we sell different snacks, including pastries, biscuits, and cooked foods for the students. We do not sell fruit, except on order, and it is usually done by the principal or teachers during special school activities. Fruits are not sold in school but can be bought outside the premises”*
(F1)


*“The school does not regulate food brought into the school sugar-sweetened beverages and pastries are available in the school premises and frequently patronised by the students”*
(P1)


*“There is no policy that enforces the limitation of unhealthy foods availability such as fried foods/snacks and carbonated drinks at the stores”*
(T3)

#### 3.1.2. A Policy That Promotes Accessibility of Healthy Foods in School Retail Stores

Stakeholders mentioned that there is no policy guiding the sales of sufficient, safe, and nutritious food at the school retail stores. Teachers primarily mentioned that food service staffs are constantly reminded of the need to keep food safe away from pests. A principal mentioned that fruits were sold for some time in the school but it was stopped due to poor patronage by the students. 


*“A programme was conducted in school to create awareness on the intake of fruits and vegetables, which prompted the school to implement a policy that ensures fruits and vegetables can be accessed by the students at the school tuck shop. The tuck shop experienced increased patronage at the time, which later dwindled. Thereafter, tuck shop had to stop sales as a consequence of poor patronage”*
(F4)


*“Despite the absence of a policy encouraging access to nutritious foods. We constantly remind the staff at the tuck shop to keep food in tight lids away from pests and they have adhered”*
(T10)


*“The school was selling fruits in the season but stopped due to poor demand by the students. The school instructed the retail storekeeper to always sell fruits in season. The patronage was encouraging at first but has gradually reduced based on poor demand”*
(T11)

#### 3.1.3. A Policy That Supports the Affordability of Healthy Foods in School

Food service staff and principal primarily mentioned that fruits especially were not affordable to the students and there was no policy in place to ensure affordability.


*“Our range of snacks such as biscuits and pastries is within the pocket money limits of the students. We do not have policy ensuring fruits are affordable, since they are not sold in the school”*
(P5)


*“There is no policy controlling prices of food sold within the school. We mostly engage our initiative, which is based on the prevailing market prices. Although the situation of things in the country has made it difficult to maintain prices as it was the previous year, we still ensure that prices are within a healthy limit. Despite our understanding that students are from healthy homes and have enough funds to purchase varieties of snacks sold”*
(F5)


*“Students usually have enough pocket money to get different things from the school tuck shop”*
(F3)

#### 3.1.4. A Policy That Supports Adolescents to Bring Fruits and Healthy Foods to School

One of the principals mentioned they have a school food policy that promotes adolescents bringing fruits and vegetables by naming the short break between 11:30 am and 12 noon a fruit break. With that, they are encouraged to bring and eat fruits. Others reported that they do not have a policy that supports healthy foods in school.


*“We have a policy that enforces fruit break between 11:30 am and 12noon to encourage more students to bring fruits from home and take it in school”*
(P6)


*“The school does not have a policy guiding bringing fruits from home. Most parents lack sufficient time to ensure varieties of healthy foods, including fruits and vegetables, are provided to their children”*
(P2)


*“Most times, adolescents are given enough cash to buy whatever food they desire from school and do not usually bring food from home, except the students in the younger class”*
(T10)

#### 3.1.5. Policy Supporting Nutrition Education Activities

Principals and teachers primarily stated that the school does not have any policy that encourages the success of nutrition edutainment activities that could promote healthy nutritional behaviour except to ensure a trained teacher in food and nutrition coordinates the activities. They all mentioned their willingness to always support any initiative that benefits the student


*“We do not have a written policy but the school usually encourages teachers to think out of the box and come up with ideas that support the nutrition education activities, the school supports fully by buying materials needed for activities”*
(P5)


*“We do not have a formal policy supporting nutrition education we occasionally invite facilitators to speak with the students on a range of moral and health-related topics. A facilitator has discussed the significance of consuming fruits. It was a one-off event”*
(T4)

#### 3.1.6. Policy for Routine Checks on Hygiene and Sanitation Within the School and Retail Store

Some principals and teachers stated that the school has a written food policy on hygiene that is enforced by employed staff whose duty is to properly clean the school facilities. The policy is well monitored and implemented. Meanwhile, others mentioned that they have a written policy on hygiene, but the school has gradually relaxed its enforcement.


*“The school has documented policy that guarantees hygienic conditions within the school environment. The school has employees that ensure the surroundings are properly cleaned. We have waste bins placed strategically at the entrance to the classes”*
(P3)


*“The school has a formal hygiene policy. It was strictly enforced during COVID-19, but currently relaxed. We used to have a hand-washing bucket at the front of each class during the [pandemic]. However, it is currently located next to the restrooms, so that students can access facilities to wash their hands after every toilet use”*
(T8)

#### 3.1.7. A Policy That Promotes Collaborations Among Staff and All Key Stakeholders to Enhance Adolescent’s Nutritional Behaviour

The majority of principals and teachers noted that while they have focused attention on several extracurricular activities, the implementation of the school food policy has been neglected. This is not because it cannot be achieved but a case of it being overlooked. The principals mentioned there is always a willingness to support staff in organising programmes and events that benefit the students. Few stated that they had been coming up with programmes such as cooking competitions.


*“There is no policy to this effect. We also cannot say we have been doing well, as regards this. Most times, the teachers take the initiative concerning nutrition programmes in the school. But, we will make it a priority to come up with several programmes at the beginning of the term that benefits and enhances lessons from their school curriculum but the enforcement of the school food policy has been unintentionally neglected”*
(P6)

### 3.2. Theme 2: Barriers and Challenges to Enforcement

#### 3.2.1. Unprioritised Funding

The principal and food service staff primarily suggested that the lack of funding makes it difficult to acquire nutritious food options, modernise kitchen equipment, train food service staff, and put nutrition education programmes into place. The type and amount of food offered may also be impacted by a lack of resources.


*“The first week of resumption is for planning activities for the term and also allocating available funds. Admittedly, nutrition programmes haven’t been a priority. This is the reason for not experiencing a significant impact. In order to have fruits and vegetables, we might have to invest some funds. There is also a need to have constant educational programmes that promote the nutrition behaviour of the student. This also requires funding, unfortunately, it hasn’t been a priority to allocate resources in its implementation at the beginning of the term”*
(P6)


*“The facilities we have at the school tuck shop are not adequate to properly store fruits and vegetables in other to keep them fresh”*
(F3)

#### 3.2.2. Resistance to Change

A teacher mentioned that the food service staff have doubts about especially making fruits available to students. This is due to poor sales experienced on their first attempt. Most food service staff primarily stated that most adolescents are not interested in buying fruits except when they are motivated to.


*“Ensuring food service staffs have access to available fresh fruits remains a challenge. They had some unpleasant experiences when we first enforced. This is because majority of the students refused to buy until it became rotten. They doubted if they could ever make adequate sales from selling healthy snacks”*
(T10)


*“Adolescents rarely requests for fruits in the tuck shop. Their demand typically determines what is sold”*
(F2)

#### 3.2.3. Competing Priorities and Time Constraints

All principals and teachers mentioned that time constraints, busy schedules, and a shortage of staff members make it difficult to prioritise school food policy monitoring tasks and devote enough time and resources to successfully adopt and maintain food policy.


*“We frequently deal with conflicting demands for resources and attention. Academic activities are at often shortened by multiple public holidays. Also, there are other extra-curricular activities the school engages in to complement the curriculum”*
(P2)

### 3.3. Theme 3: Recommendation for Successful Future Implementation of School Food Policy

#### 3.3.1. Regular Training and Development of School Food Policy Implementation Activity Manual

All principals primarily mentioned the need for constant training and monitoring for both teachers and food service staff for effective policy implementation. They encouraged the adoption of a standardised training manual that contains the roles and responsibilities of key stakeholders and initiatives schools can engage in to promote implementation, including nutrition education activities, how to ensure food is adequate in nutrients, and others.


*“There is a need for a comprehensive and standardised training manual that encourages adolescents to adopt healthy eating habits. It should include fun and educative programmes and activities”*
(P4)


*“Regular training should be provided for teachers and food service staff to ensure they are updated on best practices, nutritional requirements, and food safety regulations”*
(P5)

#### 3.3.2. Collaborative Approach

Most teachers mentioned that an implementation strategy is required. This should be a collaborative approach from the principal, teacher, and food service staff if implementation will be effective.


*“The development and implementation of a school food policy should involve principals, teachers, students, and food service personnel. All parties concerned could be encouraged to support and cooperate with the policy by acknowledging their suggestions, addressing any concerns, and outlining its advantages”*
(T10)


*“To find fresh, wholesome, and locally produced foods, there is a need to establish partnerships with local farmers and suppliers that can easily bring the perishable foods, especially fruits to school. This will reduce logistics cost and guarantees the availability and accessibility and also the affordability of nutritious through sustainable supply at a cost-effective rate”*
(F3)

#### 3.3.3. Promote Availability and Accessibility of Seasonal Foods

Both the principal and food service staff mentioned that fruits in season are fresher, higher in nutrients, and usually cheaper. 


*“Increasing the amount of seasonal fruits sold in schools is crucial. When in season, they are typically more affordable, fresher, and more nutrient-dense”*
(F2)


*“We need to pay more attention to what the food service staff stocks up in the tuck shop. There is a need to ensure that our school offers students access to a wide range of nutritious foods, such as fresh fruits, and vegetables, most especially fruits in season”*
(P2)

#### 3.3.4. Establish Efficient Communication Channels for Nutrition Education

This could be through posters and the development of other strategic nutrition education initiatives to promote the education of not only the adolescents but all stakeholders involved, including teachers, principals, and school tuck shop staff. There is a need to include nutrition education in the curriculum in other to understand the significance of eating healthy meals and the long-term benefits of nutrition behaviour. Informational resources and visual cues within the school can positively influence their decisions.


*“The development and implementation of a school food policy should involve principals, teachers, students, and food service personnel. All parties concerned could be encouraged to support and cooperate with the policy by acknowledging their suggestions, addressing any concerns, and outlining its advantages”*
(T10)

*“More emphasis should be paid to activities that can teach students to choose healthy foods. Those activities can be class competitions, debates, assembly discussions, and others. But they must be interesting, so it can be easily accepted*”(T9)


*“Adolescents must be taught to make healthy food choices. When they crave healthy foods, it will in turn drive the demand in the school tuck shop”*
(T7)


*“We need to create an attractive healthy eating environment in the school. Nutrition education should be done attractively and entertainingly. This could be through the use of posters and fliers that can promote healthy eating practices. These can improve the awareness and visibility of healthy food alternatives in the school cafeteria. Tactics to persuade students to choose healthier foods”*
(P1)

#### 3.3.5. Strong Leadership and Support

All principals mentioned that schools need to provide effective leadership in order to ensure implementation is enforced.


*“Starting from the beginning of school academic activities, school food policy’s implementation should be accorded top priority. In addition to providing financing, training of food service employees to serve nutrient-dense food, and promoting nutrition education programmes, we must continue to maintain track and enforce student compliance”*
(P4)


*“We can include reward during price giving day, on the most compliance. We need to constantly keep tabs on the application of food policies, and gather stakeholder input. Regular review aids in identifying accomplishments and potential improvement areas”*
(P5)

## 4. Discussion

In this research, we examined the perspectives of school principals, teachers, and food service staff as related to the school food policies implementation and enforcement in school. The strength of this study was our insight from the principals, teachers, and food service staff who are the major stakeholders involved in decision making regarding the school food policies. The study aims to provide a comprehensive understanding of the current state of school food policy implementation among adolescents in secondary schools. The study explored the effectiveness of existing food policies, identified barriers and challenges to the successful enforcement of school food policy, and provided recommendations for future interventions by gathering recommendations from these critical stakeholders.

The study explored the implementation of available policies that support adolescents regarding the availability, accessibility, and affordability of healthy foods in school; support nutrition education activities; support routine checks on hygiene and sanitation within the school and retail store; and promote collaborations among staff and all key stakeholders. This study observed that a higher number of schools did not implement school food policies. Only a few schools had written policies and enforcement with hygiene policies and policies involving bringing fruits to school. This confirms the association between the exposure of adolescents to food outlets in the school environment and its effect on their diet quality [35]. This is consistent with a finding showing that students who ate at the school store in private school and ate more than three times a week were overweight. These researchers also observed that high-fat, high-salt, and high-sugar foods brought more sales to the store [36]. The food policy at school is thought to have a significant impact on how adolescents who are in school choose to eat. Adolescents spend the majority of their time in school; thus, the environment and rules around school food are likely to have an impact on them [37].

Previous findings have highlighted poor school food policy implementation among schools in Nigeria [1,27,29]. However, this study emphasised each school’s individual food policy in order to understand the availability and extent of implementation from the perspectives of the key stakeholders, investigating the current state of the effectiveness of each of the major school food policies in secondary school. The report showed that most schools do not have written food policies and therefore do not implement school food policies. This is consistent with a finding in India that also reported a lack of written food policies in private schools [38]. Also, our study stated that in the few schools with written food policies, it was not enforced. This is consistent with a study by [38], who stated that 87.1% of principals said their school had a written food policy, although the majority of the time, the guidelines were not clear. He concluded that defining food regulations, improving the monitoring of them, and establishing ways to enforce them could all help to improve the school’s food policy. Numerous researchers have evaluated how school food policy implementation influences adolescents’ nutritional behaviour [39,40,41,42]. A study reported that carefully designed policies can impact students’ dietary decisions and enhance their overall nutrition [43]. These policies should include the accessibility, availability, and affordability of healthy food options. Another study reported that comprehensive school food policies can favourably influence the foods and drinks provided on the school premises and can be highly acceptable to key stakeholders without adversely affecting profitability [30].

Evidence suggests that nutrition behaviour is influenced by school food policies whether they are mandated or optional. According to a study, competitive food/beverage standards decreased the consumption of sugar-sweetened beverages by 0.18 servings per day and unhealthy snack consumption by 0.17 servings per day, and school meal standards (mostly lunch) increased the consumption of fruit by 0.76 servings per day [42]. 

This study supports previous findings reporting that the majority of the food available at school tuck shops is energy-dense, nutrient-poor, and sugar-sweetened, indicating the poor implementation of school food policies [8,44]. It also yielded a key emphasis on several other barriers and challenges hindering the enforcement of school food policies among adolescents in secondary school. Some of the challenges stressed include that implementing these policies is not prioritised within their funding. This is a situation where the school does not allocate funding for healthy food availability because they do not see a need, contrasting previous studies that reported a lack of financial resources as a barrier to schools offering a wider selection of healthy foods [45,46]. The study also suggested that inexperienced staff, resistance from food service staff and teachers, and the unavailability of a comprehensive school food club manual that serves as a guide, indicating the various roles and responsibilities of key stakeholders, contribute to barriers hindering school food policy implementation. While it supports previous studies, this study provides a robust understanding of the intricacies of the challenge. It is essential to recognise these obstacles to eliminate them and enhance policy implementation. The challenges demonstrated the interplay of barriers to the success of the school food policies. The study also agrees with a previous report that mentioned lack of training as a major setback. For instance, a report by [47] stated that almost all key informants agreed that a lack of coordination between stakeholders and limited training opportunities are the barriers to school food policy implementation. The study highlighted the need for a comprehensive school food manual that includes healthy food activities that adolescents can engage in for nutrition education and also indicates the roles and responsibilities of principals, teachers, and food service staff for use in training in order to promote successful implementation and enforcement of the policy. 

Participants emphasised that the absence of a school food activity manual has impeded the successful implementation of the school food policy. Some of the impacts include a lack of creativity in nutrition education initiatives, improper allocation of funds for nutrition education activities, and the unavailability of nutritious meals, especially seasonal fruits and vegetables. The study demonstrated how an increase in knowledge through nutrition education activities can promote nutritional behaviour that in turn drives demand for nutritious foods in the school tuck shop. This is consistent with the finding by [47], who reported that increased nutritional knowledge scores were associated with a significantly higher consumption of some healthy foods and significantly lower consumption of other unhealthy foods, according to a study of Lebanese teenagers enrolled in a private school [48]. An activity manual is suggested for use in the training of food service staff to ensure food safety and the affordability of nutritious foods. The study further emphasised that the school food manual can be adopted by schools in order to encourage unique experiences for adolescents. The successful implementation of the use of the school food activities manual has the potential to strengthen the enforcement of the existing policy, improve nutrition programme funding, enhance nutrition education initiatives, and include key stakeholders in the decision-making process.

Participants’ recommendations for effective implementation include regular training, the development of a school food policy implementation activity manual, a collaborative approach promoting the availability and accessibility of seasonal fruits, establishing efficient communication channels for nutrition education, and strong leadership and support, which are consistent with previous studies that mentioned facilitators of school food policy implementation [40,48]. 

The study’s strengths include its wide range of perspectives from important stakeholders and the rich information it obtained as a result of its qualitative methodology. We should also be aware of our study’s main limitations. The study’s small, convenient sample means that its results may not be generalisable to Nigeria. The results should be interpreted with consideration of the limitations of this qualitative research. First off, because interviewees were only chosen from Ibadan, Oyo State, the study sample could not be typical of Nigerian schools as a whole. Second, the study’s restriction to private secondary schools merely reduces how broadly the results can be applied. Nonetheless, it should be highlighted that compared to their public school peers, students at private schools have a higher prevalence of overweight and obesity [12,14], indicating the need for research in private secondary schools.

Therefore, future research should examine the perspectives of school stakeholders from different states in Nigeria and a variety of public secondary schools to improve the generalisability of the current findings.

Future studies could duplicate our methodology in a variety of geographical settings, including more remote ones, to improve and broaden the suggestions from the current study. Our qualitative data only include viewpoints from important private school stakeholders, which is another restriction. Public schools might be included in future research. Despite these drawbacks, our findings offer encouraging suggestions for future programmes to take into account, such as the development of an activity manual to serve as a guide for school policy implementation and enforcement and also the need to encourage leadership support.

## 5. Conclusions

The study reported the current situation of school food policy implementation among adolescents in private secondary schools in Nigeria. It explored the availability of healthy eating policies. It also highlighted how the school food policies are communicated, monitored, and enforced within the school setting by examining the role of school administrators, teachers, and food service staff in promoting and adhering to these policies. The study reported that most schools do not have written school food policies, and only a few have enforced them. The two major policies enforced by a few schools were a policy on hygiene and a policy that encourages adolescents to bring fruits to school. Barriers impacting poor policy implementation and enforcement include non-prioritised funding, resistance to change, and time constraints. By identifying these factors, researchers and policymakers can develop effective interventions and policies to promote healthier eating behaviors among this specific group of adolescents

Our recommendations include the regular training and development of an activity manual, promotion of seasonal foods, nutrition education, and strong leadership support. These recommendations provide insight into the challenges faced in implementing food policies by identifying viable solutions. These reports can be used to inform future interventions and policy adjustments to encourage the proper implementation and enforcement of school food policies and, in turn, encourage improved nutritional behaviour among secondary school students.

## Figures and Tables

**Table 1 ijerph-22-00866-t001:** Key informants and the number of interviews.

S/N	Key Informant	No of Interview
1	Principal	6
2.	Teachers	12
3.	Food service staff	6

**Table 2 ijerph-22-00866-t002:** Showing themes, definitions, and sample quotes.

Themes	Subthemes	Definition	Sample Quotes
Availability of school food policy	A policy that restricts the availability of unhealthy foods	Adequate nutritious food is present in the school tuck shop for sale to adolescents	“We have a written policy on the availability of healthy foods in school, but it is not enforced. Also, we sell different snacks, including pastries, biscuits, and cooked foods for the students. We do not sell fruit, except on order, and it is usually done by the principal or teachers during special school activities. Fruits are not sold in school but can be bought outside the premises” (F1). “We do not have a written food policy in the school. Also, cooked foods are not often sold in school except occasionally, when some vendors bringing in seasonal vegetables such as corn and mangoes” (F3).
	A policy that promotes accessibility of healthy foods in school retail stores	Sufficient, safe, and nutritious food that meets adolescent’s preferences and style	“A programme was conducted in school to create awareness on the intake of fruits and vegetables, which prompted the school to implement a policy that ensures fruits and vegetables can be accessed by the students at the school tuck shop. The tuck shop experienced increased patronage at the time, which later dwindled. Thereafter, tuck shop had to stop sales as a consequence of poor patronage” (F4). “Despite the absence of a policy encouraging access to nutritious foods,” We constantly remind the staff at the tuck shop to keep food in tight lids away from pests and they have adhered” (T10).
	A policy that supports the affordability of healthy foods in school	Food is not costly, considering the ratio of food price to adolescent’s pocket money	“Our range of snacks such as biscuits and pastries is within the pocket money limits of the students. We do not have policy ensuring fruits are affordable, since they are not sold in the school” (P5). “There is no policy controlling prices of food sold within the school. We mostly engage our initiative, which is based on the prevailing market prices. Although the situation of things in the country has made it difficult to maintain prices as it was the previous year, we still ensure that prices are within a healthy limit. Despite our understanding that students are from healthy homes and have enough funds to purchase varieties of snacks sold” (F5).
Availability of school food policy	A policy that encourages adolescents to bring fruits and healthy foods to school	Initiatives that encourage adolescents to bring food in lunch packs to school	“We have a policy that enforces fruit break between 11:30 am and 12noon to encourage more students to bring fruits from home and take it in school” (P6). “The school does not have a policy guiding bringing fruits from home. Most parents lack sufficient time to ensure varieties of healthy foods, including fruits and vegetables, are provided to their children” (P2).
	Policy supporting nutrition education activities	Provide awareness and motivation for improved nutrition and lifestyle	“We do not have a formal policy supporting nutrition education we occasionally invite facilitators to speak with the students on a range of moral and health-related topics. A facilitator has discussed the significance of consuming fruits. It was a one-off event” (T4)
	Policy for routine checks on hygiene and sanitation within the school and retail store		“The school has documented policy that guarantees hygienic conditions within the school environment. The school has employees that ensure the surroundings are properly cleaned. We have waste bins placed strategically at the entrance to the classes” (P3). “The school has a formal hygiene policy. It was strictly enforced during COVID-19, but currently relaxed. We used to have a hand-washing bucket at the front of each class during the [pandemic]. However, it is currently located next to the restrooms, so that students can access facilities to wash their hands after every toilet use” (T8).
	A policy that promotes collaborations among staff and all key stakeholders to enhance adolescent’s nutritional behaviour	The strategic partnership of principal, teachers, and food service staff to improve nutrition behaviour among adolescents	“The lack of a policy in this regard signifies that we are not considered to have been successful in this regard either. Teachers typically take the lead in implementing nutrition programmes in the classroom. However we are going to emphasise developing a number of initiatives at the commencement of every academic year to support and enhance the lessons taught in the classroom. However, the implementation of the school food policy has inadvertently been neglected”(P6).
Barriers to enforcement	Unprioritised funding	Funding is available but not prioritised to promote school food policies	“The first week of resumption is for planning activities for the term and also allocating available funds. Admittedly, nutrition programmes haven’t been a priority. This is the reason for not experiencing a significant impact. In order to have fruits and vegetables, we might have to invest some funds. There is also a need to have constant educational programmes that promote the nutrition behaviour of the student. This also requires funding, unfortunately, it hasn’t been a priority to allocate resources in its implementation at the beginning of the term” (P6).
	Resistance to change	The implementation of the food policy has been hampered by opposition from students, some teachers, and food service staff. Personal preferences, cultural standards, and doubts about the viability and usefulness of the changes are major sources of resistance	Ensuring food service staffs have access to available fresh fruits remains a challenge. They had some unpleasant experiences when we first enforced. This is because majority of the students refused to buy until it became rotten. They doubted if they could ever make adequate sales from selling healthy snacks (T10). “Adolescents rarely requests for fruits in the tuck shop. Their demand typically determines what is sold” (F2).
	Competing priorities and time constraints	Healthy eating requires time, resources, and knowledge, which might be limited	“We frequently deal with conflicting demands for resources and attention. Academic activities are at often shortened by multiple public holidays. Also, there are other extra-curricular activities the school engages in to complement the curriculum” (P2).
Recommendations	Regular training and development of school food policy implementation activity manual	Need to train teachers and food service staff on how to ensure safe and adequate nutritious food is available in the school at all times	“There is a need for a comprehensive and standardised training manual that encourages adolescents to adopt healthy eating habits. It should include fun and educative programmes and activities” (P4).
	Collaborative approach	All stakeholders join hands in the implementation and enforcement of the school food policy	“The development and implementation of a school food policy should involve principals, teachers, students, and food service personnel. All parties concerned could be encouraged to support and cooperate with the policy by acknowledging their suggestions, addressing any concerns, and outlining its advantages” (T10).
	Promote the availability and accessibility of seasonal foods		“Increasing the amount of seasonal fruits sold in schools is crucial. When in season, they are typically more affordable, fresher, and more nutrient-dense” (F2).
	Establish efficient communication channels for nutrition education	Strategic nutrition education initiatives to promote the education of not only adolescents but all stakeholders	“Just as there are other health posters in the schools, there should be posters teaching students how to eat healthily” (T10). “Activities that assist students adopt healthy food choices deserve more attention. These can include debates, nutrition education on assembly ground, class competitions, and more. However, they have to be captivating in order for it to be readily accepted” (T13).
	Effective leadership to support and enforce school food policy implementation in school		“Starting from the beginning of school academic activities, school food policy’s implementation should be accorded top priority. In addition to providing financing, training of food service employees to serve nutrient-dense food, and promoting nutrition education programmes, we must continue to maintain track and enforce student compliance” (P4).

## Data Availability

Data supporting results are available on request.

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
