# Peer review of "Assessment of School Food Policy Influencing Nutritional Behaviour of Adolescents from the Perspective of School Stakeholders in Ibadan, Oyo State"

_ijerph, 2025, doi:10.3390/ijerph22060866_

Round 1

Reviewer 1 Report

Comments and Suggestions for Authors

Dear Authors, I know how disappointing it can be to receive reviews that require further work, but it is a reviewer’s duty to contribute to high-quality work.

Abstract: The abstract provides a clear overview of the study's aims, methodology, and findings. It effectively communicates the importance of exploring the perspectives of key stakeholders in the implementation of school food policy.

*Ensure consistency in the terminology used throughout the paper. For example, "other to" in line 124 should be corrected to "in order to."

Introduction: The suggestion here is to provide more context on the existing school food policies and their implementation status. This could help readers understand the specific challenges and gaps in policy enforcement that contribute to the prevalence of unhealthy food choices in schools. Furthermore, clearly stating the objectives of the study in this section will help to guide readers through the research aims and goals.

Materials and Methods: This section provides a detailed description of the qualitative study design and data collection process. Here are a few comments and suggestions for improvement:

-It is important to clarify the rationale behind choosing private English- speaking secondary schools in Ibadan, Oyo State, as the study areas. Providing justification for this selection will enhance the understanding of the study context and its relevance to the research objectives.

-Specify the criteria used for selecting the researchers who conducted the interviews. Providing information on their qualifications and expertise in qualitative interviewing methods would add credibility to the study.

Discussion: This section appropriately acknowledges the limitations of the study, such as the small sample size and the focus on private schools, which may limit the generalizability of the findings. To enhance transparency, it would be helpful to discuss how these limitations may have influenced the study results and interpretations.

Author Response

 Comments 1: The abstract provides a clear overview of the study's aims, methodology, and findings. It effectively communicates the importance of exploring the perspectives of key stakeholders in the implementation of school food policy.

*Ensure consistency in the terminology used throughout the paper. For example, "other to" in line 124 should be corrected to "in order to."

Response 1:  Thank you for pointing this out. I/We agree with this comment. Therefore, I/we have changed it to “Therefore, to “- Line  per section (68), Page 3 0f 20, Paragraph per section (6) 

Comments 2: Introduction: The suggestion here is to provide more context on the existing school food policies and their implementation status. This could help readers understand the specific challenges and gaps in policy enforcement that contribute to the prevalence of unhealthy food choices in schools. Furthermore, clearly stating the objectives of the study in this section will help to guide readers through the research aims and goals.

Response 2: Agree. I/We have, accordingly, done/revised/changed/modified as below.

In addition, Nigeria introduced a National school feeding programme in  2016, to provide adequate and nutritious meals to all primary school pupils (1,25). The programme has been reported to take place in 26 states around the country, involving 80,000 farmers and over 102,097 cooks with over nine million children benefiting from 54,619 schools (26). Studies have shown that participants are aware of the existence of the programme, but have reported poor implementation(1,27). These studies have shown poor quality and quantity of meals provided for the pupils. Also, the school food policy emphasizes on incorporating nutrition education and training into curricula and encouraging healthy eating practices. These have also faced challenges in effective implementation. Thereby contributing to the increase in poor nutritional behaviour among adolescents. Some of the challenges resulting in poor implementation as reported include inadequate resources, poor monitoring and poor quality of school meals, and limited training opportunities (1,27–29).- Line per section ( 47-58); Paragraph per section (4), page 3 of 20

Therefore, the purpose of this study is to understand the perspectives of school principals, teachers, and food service staff as related to the school food policies implementation and enforcement in school. Line per section (85-87); Paragraph per section (7), Page 3 0f 20.

 Comment 3: Materials and Methods: This section provides a detailed description of the qualitative study design and data collection process. Here are a few comments and suggestions for improvement:

-It is important to clarify the rationale behind choosing private English- speaking secondary schools in Ibadan, Oyo State, as the study areas. Providing justification for this selection will enhance the understanding of the study context and its relevance to the research objectives.

-Specify the criteria used for selecting the researchers who conducted the interviews. Providing information on their qualifications and expertise in qualitative interviewing methods would add credibility to the study.

Response 3: Agree. I/We have, accordingly, done/revised/changed/modified:

Private school students are mostly from the high or middle-income class with reported cases of non-communicable diseases (31). According to Aa and Ahmed's (2014) study, adolescents in this socioeconomic rung of the social scale had a greater prevalence of obesity and overweight. Therefore, targeted intervention for this group of young people will aim at limiting the grievous consequences of non-communicable diseases. From the list of private secondary schools, six schools were randomly selected. Location includes Sango, Oluyole, Bodija, Mokola, Agbowo, and Total Garden. All within the Ibadan metropolis, Oyo State.Line per section: 11-18,  Paragraph per section: 1, Page:3 of 20

Five research assistants who were graduates with qualitative research experience, two from the field of public health and three from the Department of Human Nutrition and Dietetics were recruited for the study. The Research Assistants further received a  two-day training. The research team was instructed to use a training manual that was designed specifically for the training activity. The training included the basics of research principles, the importance of objectivity and unbias, the research objectives, and lastly, how to implement KII and FGD sessions.

 Line per section:4-10  Paragraph per section:1  Page:3 of 20

Comment 4: Discussion: This section appropriately acknowledges the limitations of the study, such as the small sample size and the focus on private schools, which may limit the generalizability of the findings. To enhance transparency, it would be helpful to discuss how these limitations may have influenced the study results and interpretations.

Response 3: Agree. I/We have, accordingly, done/revised/changed/modified:

The results should be interpreted with consideration for the limitations of this qualitative research. First off, because interviewees were only chosen from Ibadan, Oyo State, the study sample could not be typical of Nigerian schools as a whole. Second, the study's restriction to private secondary schools merely reduces how broadly the results can be applied. Nonetheless, it should be highlighted that compared to their public school peers, students at private schools have a higher prevalence of overweight and obesity (12,14), indicating the need for research in private secondary schools.

 Therefore, future research should examine the perspectives of school stakeholders from different states in Nigeria and a variety of public secondary schools to improve the generalizability of the current findings.

 Line per section:96-104  Paragraph per section:8 , Page:17 of 20

Reviewer 2 Report

Comments and Suggestions for Authors

Dear authors,

This is not a new situation. Additionally, the number of participants is very low.

Comments on the Quality of English Language

there are many grammatical errors.

Author Response

Response to Reviewer 2 Comments

1. Summary

Comments 1: This is not a new situation. Additionally, the number of participants is very low.

Response 1: Thank you for pointing this out. However, from literature review, There is no known study reviewing the perspectives of stakeholders as related to the school food policies implementation and enforcement in school in Nigeria. These  group of people have been reported to be at high risk of non-communicable diseases. Therefore, this study aims at closing this gap.

Also, the participants number was reached when the data collected became saturated as stated in the method section.

The researcher ensured saturation was achieved across the interviewees by checking for no new data and redundant data for each of the specific questions asked. After approximately 24 interviews, the researcher determined that saturation occurred, and thus, no additional interviews were needed.

Line per section: 57-60 Paragraph per section: 6, Page:4 of 20

Comments 2: Comments on the Quality of English Language, there are many grammatical errors.

Response 2: Agree. I/We have, accordingly, done/revised/changed/modified. Quality of English has been improved

Reviewer 3 Report

Comments and Suggestions for Authors

Dear Authors,

Thank you for the opportunity given to me to review the article entitled Assessment of School "Food Policy Influencing Nutritional Behavior of Adolescents from the Perspective of School Stakeholders in Ibadan, Oyo State". Articles submitted and proposed to IJERPH are worthy enough to be published by MDPI. I also positively welcome the writing that is presented in a positive way. However, before entering the consideration phase by the Editorial Board, several suggestions are recommended in this paper as an effort to improve the quality of the writing. There are five comments that require correction from the author's colleagues as follows:

·         Q1. The sample was selected from six private secondary schools. Can the authors detail the list of schools? So, where exactly was this interview conducted? I wish the location should be mentioned in the abstract.

·         Q2. The contribution of this paper is only intended for practical improvements. However, I have not seen the expectations for the sustainability of similar studies in the future. Please add arguments that can strengthen the output of the paper for theoretical and academic strengthening at the end of the paragraph (Introductory Chapter).

·         Q3. Weaknesses in the Discussion. This section not only presents findings based on field interviews, but also compares them with past studies. Therefore, I suggest you expand your references (at least 5-6 publications from 2020).

·         Q4. The conclusion is also weak. Generally, conclusions not only confirm existing findings and managerial policies, but also look at study limitations to produce new recommendations for the direction of further research development.

·         Q5. Is this citing technique justified? As far as I know, MDPI has its own citation style. I hope that the author's colleagues can review the standard guidelines, especially writing references, which are implemented by MDPI. You can also study previous articles published by MDPI.

Finally, I submitted the results of the first review as author notes. I await responses from the authors through the latest revisions.

Sirecenly,

Anonymous reviewer

Author Response

Response to Reviewer 3 Comments

1. Summary

Comment 1: Q1. The sample was selected from six private secondary schools. Can the authors detail the list of schools? So, where exactly was this interview conducted? I wish the location should be mentioned in the abstract.

Response1: Thank you for pointing this out. However, I will not be able to list the names the schools, because it will result in a bridge of confidentiality. Also, due to the constraint in number of words in abstract, the location of the schools has been included in the method section.

From the list of private secondary schools, six schools were randomly selected. Location includes, Sango, Oluyole, Bodija, Mokola, Agbowo and Total garden. All within Ibadan metropolis, Oyo State.

Line per section: 16-18  Paragraph per section: 1 Page: 1

Comment 2: Q2. The contribution of this paper is only intended for practical improvements. However, I have not seen the expectations for the sustainability of similar studies in the future. Please add arguments that can strengthen the output of the paper for theoretical and academic strengthening at the end of the paragraph (Introductory Chapter).

Response2: Thank you for pointing this out. I believe this aspect of sustainability is covered in the last part of the introductory section as coated below:

“By examining the role of school administrators, teachers, and food service staff in promoting and adhering to the policies. The opinions of major stakeholders can shed insight into the challenges of implementing food policies and aid in the identification of potential solutions. These findings can be used to guide future interventions and policy changes to promote the proper implementation of school food policies, thereby promoting healthier nutritional behaviour among secondary school students”.

Line per section: 79-85, Paragraph per section: 7, Page: 3 of 20

Comment 3: Q3. Weaknesses in the Discussion. This section not only presents findings based on field interviews, but also compares them with past studies. Therefore, I suggest you expand your references (at least 5-6 publications from 2020).

Response3: Thank you for pointing this out. References has been expanded to include seven (7) more publications. Including: (1,27,29); (8,44); (45,46).

Line per section: 26 ,Paragraph per section:3 , Page: 15 of 20

Line per section:  Paragraph per section:  Page:

Line per section:  Paragraph per section:  Page:

Comment 4: Q4. The conclusion is also weak. Generally, conclusions not only confirm existing findings and managerial policies, but also look at study limitations to produce new recommendations for the direction of further research development.

Response 4: Thank you for pointing this out. However, I realized that MDPI articles include limitation in the discussion section. Therefore, in other to bridge the gap mentioned, limitations and recommendations has now been included in the discussion section. The conclusion section also contains focus for researches and programme implementation.

Line per section:51,  Paragraph per section: 5, Page:16 of 20

Comment 5: Q5. Is this citing technique justified? As far as I know, MDPI has its own citation style. I hope that the author's colleagues can review the standard guidelines, especially writing references, which are implemented by MDPI. You can also study previous articles published by MDPI.

Response5: The MDPI referencing style has been implemented

Reviewer 4 Report

Comments and Suggestions for Authors

Dear Author, 

1. your work on behavior science that need Flow diagram of selected data for graphical abstract. 

2. The reason for new school food policy need to emphasize in abstract and introduction. 

3.  The school food policy (SFP) must be standardized with national bench marking.

4.  Line 112 in "The qualitative study utilized a thematic analysis" need to clarify. 

5. The line 171 need to clear in segment wise.

6. conclusion must be improved. 

7. This paper is on behavior science and not meet the thrust areas of this journal.

Comments on the Quality of English Language

Extensive editing of English language need to incorporate.

Author Response

Response to Reviewer 4 Comments

1. Summary

.

Comment 1: your work on behavior science that need Flow diagram of selected data for graphical abstract. 

Response 1: Thank you for pointing this out. However, the study can be grouped under chronic diseases and prevention and not behavioural science. Which I believe is very much within the boundaries of this journal. The study therefore does not require flow diagram.

 Comment 2: The reason for new school food policy need to emphasize in abstract and introduction. 

Response 2: Thank you for pointing this out. However, the study does not necessarily project the need for new school food policy but an understanding of the opinions of major stakeholders pertaining the current implementation of the school food policy, how it can provide insight into the challenges of implementing food policies and aid in the identification of potential solutions to promote effective implementation. This was adequately represented in the abstract and introduction as below.

Abstract “The study provides a deeper understanding of the role of key stakeholders in the current state of the effectiveness of SFP implementation. The inclusion of stakeholders is essential for a successful school food policy”.

Line per section:13-16,  Paragraph per section: 1, Page: 1 of 20

Introduction “The study's major findings will present an overview of the current situation of school food policy implementation, exploring the availability of healthy eating policies. It will also highlight how these policies are communicated, monitored, and enforced within the school setting. By examining the role of school administrators, teachers, and food service staff in promoting and adhering to the policies. The opinions of major stakeholders can shed insight into the challenges of implementing food policies and aid in the identification of potential solutions. These findings can be used to guide future interventions and policy changes to promote the proper implementation of school food policies, thereby promoting healthier nutritional behaviour among secondary school students”.

Line per section: 77-85, Paragraph per section: 7, Page:3 of 20

Comment 3: The school food policy (SFP) must be standardized with national bench marking.

Response 3: Thank you for pointing this out. However, the study will not be introducing a new school food policy and therefore does not require standardising with national bench marking.

Comment 4:  Line 112 in "The qualitative study utilized a thematic analysis" need to clarify. 

Response 4: Thank you for pointing this out. The qualitative data analysis method requires identifying pattern across data to uncover themes. In this study, some of the themes derived includes; Availability of School food policy, Barriers to enforcement and recommendations.

Comment 5:  The line 171 need to clear in segment wise.

Response 5: Thank you for pointing this out. I believe the section appropriately outlined the stages

The objective of the study was explained and informed consent was obtained.  The discussants requested approval to utilise a digitised recording device before the KII sessions started. All identifiers were erased, including participant names. The numbers as-signed to each person were not connected to any identifiers and were just used to facilitate discussion. Both verbal and nonverbal communications were also recorded by an observer. Yin's five-phase analysis method, which entails compiling, disassembling, re-assembling, interpreting, and concluding, was employed in analysing the data (34)

Line per section: 61-67, Paragraph per section:7,  Page: 4 of 20

Comment 6: conclusion must be improved. 

Response 6: Thank you for pointing this out. However, In some journals, conclusion includes limitations. But going through MDPI articles, limitations are written under discussions. This guideline was followed and the discussion section has been improved.

The results should be interpreted with consideration for the limitations of this qualitative research. First off, because interviewees were only chosen from Ibadan, Oyo State, the study sample could not be typical of Nigerian schools as a whole. Second, the study's restriction to private secondary schools merely reduces how broadly the results can be applied. Nonetheless, it should be highlighted that compared to their public school peers, students at private schools have a higher prevalence of overweight and obesity (12,14), indicating the need for research in private secondary schools.

 Therefore, future research should examine the perspectives of school stakeholders from different states in Nigeria and a variety of public secondary schools to improve the generalizability of the current findings.

Line per section: 96-104,  Paragraph per section: 8, Page: 17 of 20

Comment 7: This paper is on behavior science and not meet the thrust areas of this journal

Response 7: Thank you for pointing this out.  However, the study can be grouped under chronic diseases and prevention. Which I believe is very much within the boundaries of this journal

Comment 8: Extensive editing of English language need to incorporporated

Response 8: Thank you for pointing this out. Quality of English has been improved

Round 2

Reviewer 2 Report

Comments and Suggestions for Authors

Dear authors,

This form is better than the first one. Thank you for your revision.

Best regards,

Comments on the Quality of English Language

Quality of English language is fine.

Author Response

Thank you very much for taking the time to review this manuscript. Please find the detailed responses attached, including the corresponding revisions in track changes in the re-submitted files.

Reviewer 4 Report

Comments and Suggestions for Authors

Dear Authors,

Your efforts to write about school food policy is impressive, only few points need to be address in revised manuscript are...

·         Kindly justify the reason to select “A thematic approach “for analysis.

·         Data is not well presented so make sub- heading and segregate the entire work to highlight the following heading.

·         Study design, setting and recruitment and ·         Data collection process.

·         Nutrition measures is not documented kindly include.

·         Sample size calculation formula and process is missing.

·         Participant characteristics and baseline measures is not written.

·         Participant recruitment flow chart should be included in material and methods.

all tests should be mentioned in results part also

Discussion needs to rewrite as per material and results.

Comments on the Quality of English Language

Extensive english correction required

Author Response

Thank you very much for taking the time to review this manuscript. Please find the detailed responses attached, including the corresponding revisions in track changes in the re-submitted files
